# Learning Delays in Spiking Neural Networks using Dilated Convolutions with Learnable Spacings

**Ilyass Hammouamri**
CerCo UMR 5549
CNRS – Université Toulouse, France
ilyass.hammouamri@cnrs.fr

**Ismail Khalfaoui-Hassani**
Artificial and Natural Intelligence Toulouse Institute (ANITI)
Université de Toulouse, France
ismail.khalfaoui-hassani@univ-tlse3.fr

**Timothée Masquelier**
CerCo UMR 5549
CNRS – Université Toulouse, France
timothee.masquelier@cnrs.fr

## Abstract

Spiking Neural Networks (SNNs) are a promising research direction for building power-efficient information processing systems, especially for temporal tasks such as speech recognition. In SNNs, delays refer to the time needed for one spike to travel from one neuron to another. These delays matter because they influence the spike arrival times, and it is well-known that spiking neurons respond more strongly to coincident input spikes. More formally, it has been shown theoretically that plastic delays greatly increase the expressivity in SNNs. Yet, efficient algorithms to learn these delays have been lacking. Here, we propose a new discrete-time algorithm that addresses this issue in deep feedforward SNNs using backpropagation, in an offline manner. To simulate delays between consecutive layers, we use 1D convolutions across time. The kernels contain only a few non-zero weights – one per synapse – whose positions correspond to the delays. These positions are learned together with the weights using the recently proposed Dilated Convolution with Learnable Spacings (DCLS). We evaluated our method on three datasets: the Spiking Heidelberg Dataset (SHD), the Spiking Speech Commands (SSC) and its non-spiking version Google Speech Commands v0.02 (GSC) benchmarks, which require detecting temporal patterns. We used feedforward SNNs with two or three hidden fully connected layers, and vanilla leaky integrate-and-fire neurons. We showed that fixed random delays help and that learning them helps even more. Furthermore, our method outperformed the state-of-the-art in the three datasets without using recurrent connections and with substantially fewer parameters. Our work demonstrates the potential of delay learning in developing accurate and precise models for temporal data processing. Our code is based on PyTorch / SpikingJelly and available at: https://github.com/Thvnvtos/SNN-delays

## 1 Introduction

Spiking neurons are coincidence detectors (König et al., 1996; Rossant et al., 2011): they respond more when receiving synchronous, rather than asynchronous, spikes. Importantly, it is the spike arrival times that should coincide, not the spike emitting times – these times are different because propagation is usually not instantaneous. There is a delay between spike emission and reception, called delay of connections, which can vary across connections. Thanks to these heterogeneous delays, neurons can detect complex spatiotemporal spike patterns, not just synchrony patterns (Izhikevich, 2006) (see Figure 1).

In the brain, the delay of a connection corresponds to the sum of the axonal, synaptic, and dendritic delays. It can reach several tens of milliseconds, but it can also be much shorter (1 ms or less) (Izhikevich, 2006). For example, the axonal delay can be reduced with myelination, which is an adaptive process that is required to learn some tasks (see Bowers (2017) for a review). In other words, learning in the brain can not be reduced to synaptic plasticity. Delay learning is also important.

A certain theoretical work has led to the same conclusion: Maass and Schmitt demonstrated, using simple spiking neuron models, that a SNN with k adjustable delays can compute a much richer class of functions than a threshold circuit with k adjustable weights (Maass & Schmitt, 1999).

Finally, on most neuromorphic chips, synapses have a programmable delay. This is the case for Intel Loihi (Davies et al., 2018), IBM TrueNorth (Akopyan et al., 2015), SpiNNaker (Furber et al., 2014) and SENeCA (Yousefzadeh et al., 2022).

All these points have motivated us and others (see related works in the next section) to propose delay learning rules. Here, we show that delays can be learned together with the weights, using backpropagation, in arbitrarily deep SNNs. More specifically, we first show that there is a mathematical equivalence between 1D temporal convolutions and connection delays. Thanks to this equivalence, we then demonstrate that the delays can be learned using Dilated Convolution with Learnable Spacings (Khalfaoui-Hassani et al., 2023a;b), which was recently proposed for another purpose, namely to increase receptive field sizes in non-spiking 2D CNNs for computer vision. In practice, the method is fully integrated with PyTorch and leverages its automatic differentiation engine.

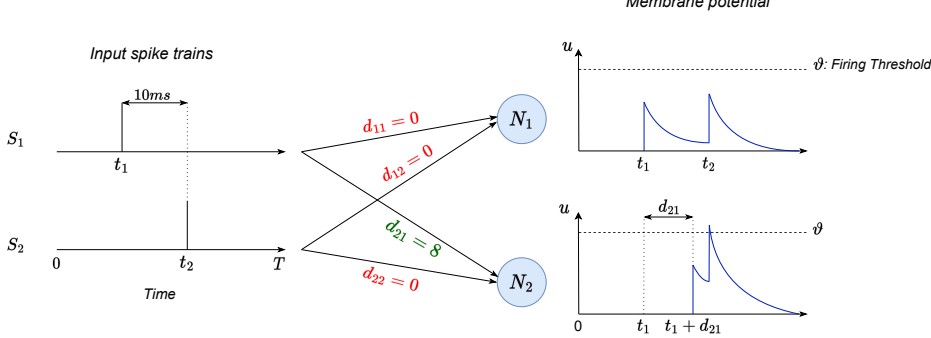

Figure 1: Coincidence detection: we consider two neurons $N_1$ and $N_2$ with the same positive synaptic weight values. $N_2$ has a delayed synaptic connection denoted $d_{21}$ of $8$ms, thus both spikes from spike train $S_1$ and $S_2$ will reach $N_2$ quasi-simultaneously. As a result, the membrane potential of $N_2$ will reach the threshold $\vartheta$ and $N_2$ will emit a spike. On the other hand, $N_1$ will not react to these same input spike trains.

## 2 RELATED WORK

### 2.1 DEEP LEARNING FOR SPIKING NEURAL NETWORKS

Recent advances in SNN training methods like the surrogate gradient method (Neftci et al., 2018; Shrestha & Orchard, 2018) and the ANN2SNN conversion methods (Bu et al., 2022; Deng & Gu, 2021; Han et al., 2020) made it possible to train increasingly deeper spiking neural networks. The surrogate gradient method defines a continuous relaxation of the non-smooth spiking nonlinearity: it replaces the gradient of the Heaviside function used in the spike-generating process with a smooth surrogate gradient that is suitable for optimization. On the other hand, the ANN2SNN methods convert conventional artificial neural networks (ANNs) into SNNs by copying the weights from ANNs while trying to minimize the conversion error.

Other works have explored improving the spiking neurons using inspiration from biological mechanisms or techniques used in ANNs. The Parametric Leaky Integrate-and-Fire (PLIF) (Fang et al., 2021a) incorporates learnable membrane time constants that could be trained jointly with synaptic weights. Bellec et al. (2018) were the first to propose a method for dynamically adapting firing

thresholds in deep (recurrent) SNNs, Hammouamri et al. (2022) also proposes a method to dynamically adapt firing thresholds in order to improve continual learning in SNNs. Spike-Element-Wise ResNet (Fang et al., 2021b) addresses the problem of vanishing/exploding gradient in the plain Spiking ResNet caused by sigmoid-like surrogate functions and successfully trained the first deep SNN with more than 150 layers. Spikformer (Zhou et al., 2023) adapts the softmax-based self-attention mechanism of Transformers (Vaswani et al., 2017) to a spike-based formulation. Other recent works like SpikeGPT (Zhu et al., 2023) and Spikingformer (Zhou et al., 2023) also proposes spike-based transformer architectures. These efforts have resulted in closing the gap between the performance of ANNs and SNNs on many widely used benchmarks.

## 2.2 DELAYS IN SNNs

Few previous works considered learning delays in SNNs. Wang et al. (2019) proposed a similar method to ours in which they convolve spike trains with an exponential kernel so that the gradient of the loss with respect to the delay can be calculated. However, their method is used only for a shallow SNN with no hidden layers.

Other methods like Grimaldi & Perrinet (2022; 2023); Zhang et al. (2020); Taherkhani et al. (2015) also proposed learning rules developed specifically for shallow SNNs with only one layer. Hazan et al. (2022) proposed to learn temporal delays with Spike Timing Dependent Plasticity (STDP) in weightless SNNs. Han et al. (2021) proposed a method for delay-weight supervised learning in optical spiking neural networks. Patiño-Saucedo et al. (2023) proposed a method for deep feedforward SNNs that uses a set of multiple fixed delayed synaptic connections for the same two neurons before pruning them depending on the magnitude of the learned weights.

To the best of our knowledge, SLAYER (Shrestha & Orchard, 2018) and Sun et al. (2022; 2023b;a) (which are based on SLAYER) are the only ones to learn delays and weights jointly in a deep SNN. However, unless a Spike Response Model (SRM) (Gerstner, 1995) is used, the gradient of the spikes with respect to the delays is numerically estimated using finite difference approximation, and we think that those gradients are not precise enough as we achieve similar performance in our experiments with fixed random delays (see Table 2 and Figure 4).

We propose a control test that was not considered by the previous works and that we deem necessary: the SNN with delay learning should outperform an equivalent SNN with fixed random and uniformly distributed delays, especially with sparse connectivity.

## 3 METHODS

### 3.1 SPIKING NEURON MODEL

The spiking neuron, which is the fundamental building block of SNNs, can be simulated using various models. In this work, we use the Leaky Integrate-and-Fire model (Gerstner & Kistler, 2002), which is the most widely used for its simplicity and efficiency. The membrane potential $u_i^{(l)}$ of the $i$-th neuron in layer $l$ follows the differential equation:

$$\tau \frac{du_i^{(l)}}{dt} = -(u_i^{(l)}(t) - u_{\text{reset}}) + RI_i^{(l)}(t) \tag{1}$$

where $\tau$ is the membrane time constant, $u_{reset}$ the potential at rest, $R$ the input resistance and $I_i^{(l)}(t)$ the input current of the neuron at time $t$. In addition to the sub-threshold dynamics, a neuron emits a unitary spike $S_i^{(l)}$ when its membrane potential exceeds the threshold $\vartheta$, after which it is instantaneously reset to $u_{reset}$. Finally, the input current $I_i^{(l)}(t)$ is stateless and represented as the sum of afferent weights $W_{ij}^{(l)}$ multiplied by spikes $S_j^{(l-1)}(t)$:

$$I_i^{(l)}(t) = \sum_j W_{ij}^{(l)} S_j^{(l-1)}(t) \tag{2}$$

We formulate the above equations in discrete time using Euler's method approximation, and using $u_{reset} = 0$ and $R = \tau$.

$$u_i^{(l)}[t] = (1 - \frac{1}{\tau})u_i^{(l)}[t-1] + I_i^{(l)}[t] \tag{3}$$

$$I_i^{(l)}[t] = \sum_j W_{ij}^{(l)} S_j^{(l-1)}[t] \tag{4}$$

$$S_i^{(l)}[t] = \Theta(u_i^l[t] - \vartheta) \tag{5}$$

We use the surrogate gradient method (Neftci et al., 2018) and define $\Theta'(x) \triangleq \sigma'(x)$ during the backward step, where $\sigma(x)$ is the surrogate arctangent function (Fang et al., 2021a).

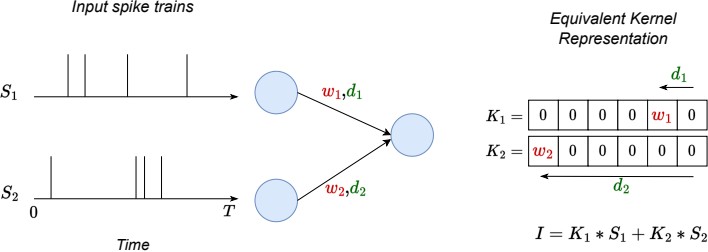

Figure 2: Example of one neuron with 2 afferent synaptic connections, convolving $K1$ and $K2$ with the zero left-padded $S_1$ and $S_2$ is equivalent to following Equation 6

## 3.2 SYNAPTIC DELAYS AS A TEMPORAL CONVOLUTION

In the following, for clarity, we assume one synapse only between pairs of neurons (modeled with a kernel containing only one non-zero element). Generalization to multiple synapses (kernels with multiple non-zero elements) is trivial and will be explored in the experiments.

A feed-forward SNN model with delays is parameterized with $W = (w_{ij}^{(l)}) \in \mathbb{R}$ and $D = (d_{ij}^{(l)}) \in \mathbb{R}^+$, where the input of neuron $i$ at layer $l$ is

$$I_i^{(l)}[t] = \sum_j w_{ij}^{(l)} S_j^{(l-1)}[t - d_{ij}^{(l)}] \tag{6}$$

We model a synaptic connection from neuron $j$ in layer $l-1$ to neuron $i$ in layer $l$ which have a synpatic weight $w_{ij}^{(l)}$ and delay $d_{ij}^{(l)}$ as a one dimensional temporal convolution (see Figure 2) with kernel $k_{ij}^{(l)}$ as follows:

$\forall n \in [\![0, ... T_d - 1]\!]$:

$$k_{ij}^{(l)}[n] = \begin{cases} w_{ij}^{(l)} & \text{if } n = T_d - d_{ij}^{(l)} - 1 \\ 0 & \text{otherwise} \end{cases} \tag{7}$$

where $T_d$ is the kernel size or maximum delay + 1. Thus we redefine the input $I_i^{(l)}$ in Equation 6 as a sum of convolutions:

$$I_i^{(l)} = \sum_j k_{ij}^{(l)} * S_j^{(l-1)} \tag{8}$$

We used a zero left-padding with size $T_d - 1$ on the input spike trains $S$ so that $I[0]$ does correspond to $t = 0$. Moreover, a zero right-padding could also be used, but it is optional; it could increase the

expressivity of the learned delays with the drawback of increasing the processing time as the number of time-steps after the convolution will increase.

To learn the kernel elements positions (i.e., delays), we use the 1D version of DCLS (Khalfaoui-Hassani et al., 2023a) with a Gaussian kernel (Khalfaoui-Hassani et al., 2023b) centered at $T_d - d_{ij}^{(l)} - 1$, where $d_{ij}^{(l)} \in [\![0,\ T_d - 1]\!]$, and of standard deviation $\sigma_{ij}^{(l)} \in \mathbb{R}^*$, thus we have:

$\forall n \in [\![0, \dots T_d - 1]\!]$:

$$k_{ij}^{(l)}[n] = \frac{w_{ij}^{(l)}}{c} \exp\left(-\frac{1}{2}\left(\frac{n - T_d + d_{ij}^{(l)} + 1}{\sigma_{ij}^{(l)}}\right)^2\right) \qquad (9)$$

With

$$c = \epsilon + \sum_{n=0}^{T_d-1} \exp\left(-\frac{1}{2}\left(\frac{n - T_d + d_{ij}^{(l)} + 1}{\sigma_{ij}^{(l)}}\right)^2\right) \qquad (10)$$

a normalization term and $\epsilon = 1e-7$ to avoid division by zero, assuming that the tensors are in `float32` precision. During training, $d_{ij}^{(l)}$ are clamped after every batch to ensure their value stays in $[\![0, \dots T_d - 1]\!]$.

The learnable parameters of the 1D DCLS layer with Gaussian interpolation are the weights $w_{ij}$, the corresponding delays $d_{ij}$, and the standard deviations $\sigma_{ij}$. However, in our case, $\sigma_{ij}$ are not learned, and all kernels in our model share the same decreasing standard deviation, which will be denoted as $\sigma$. Throughout training, we exponentially decrease $\sigma$ as our end goal is to have a sparse kernel where only the delay position is non-zero and corresponds to the weight.

The Gaussian kernel transforms the discrete positions of the delays into a smoother kernel (see Figure 5), which enables the calculation of the gradients $\frac{\partial L}{\partial d_{ij}^{(l)}}$.

By adjusting the parameter $\sigma$, we can regulate the temporal scale of the dependencies. A small value for $\sigma$ enables the capturing of variations that occur within a brief time frame. In contrast, a larger value of $\sigma$ facilitates the detection of temporal dependencies that extend over longer durations. Thus, $\sigma$ tuning is crucial to the trade-off between short-term precision and long-term dependencies.

We start with a high $\sigma$ value and exponentially reduce it throughout the training process, after each epoch, until it reaches its minimum value of 0.5 (Fig. 3). This approach facilitates the learning of distant long-term dependencies at the initial time. Subsequently, when $\sigma$ has a smaller value, it enables refining both weights and delays with more precision, making the Gaussian kernel more similar to the discrete kernel that is used at inference time. As we will see later in our ablation study (Section 4.3), this approach outperforms a constant $\sigma$.

Indeed, the Gaussian kernel is only used to train the model; when evaluating on the validation or test set, it is converted to a discrete kernel as described in Equation 7 by rounding the delays. This permits to implement sparse kernels for inference which are very useful for uses on neuromorphic hardware, for example, as they correspond to only one synapse between pairs of neurons, with the corresponding weight and delay.

## 4 EXPERIMENTS

### 4.1 EXPERIMENTAL SETUP

We chose to evaluate our method on the SHD (Spiking Heidelberg Digits) and SSC (Spiking Speech Commands)/GSC (Google Speech Commands v0.02) datasets (Cramer et al., 2022), as they require leveraging temporal patterns of spike times to achieve a good classification accuracy, unlike most computer vision spiking benchmarks. Both spiking datasets are constructed using artificial cochlear models to convert audio speech data to spikes; the original audio datasets are the Heidelberg Dataset (HD) and the GSC v0.02 Dataset (SC) (Warden, 2018) for SHD and SSC, respectively.

The SHD dataset consists of 10k recordings of 20 different classes that consist of spoken digits ranging from zero to nine in both English and German languages. SSC and GSC are much larger

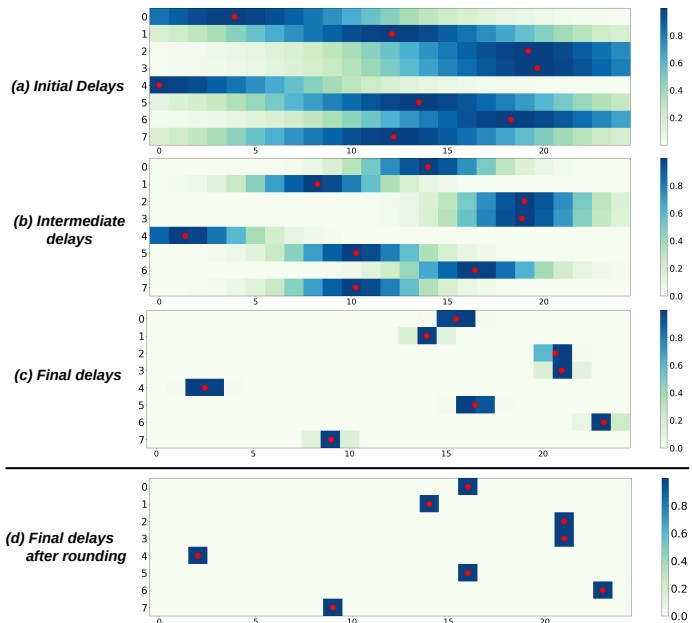

Figure 3: This figure illustrates the evolution of the same delay kernels for an example of eight synaptic connections of one neuron throughout the training process. The x-axis corresponds to time, and each kernel is of size $T_d = 25$. And the y-axis is the synapse id. (a) corresponds to the initial phase where the standard deviation of the Gaussian $\sigma$ is large ($\frac{T_d}{2}$), allowing to take into consideration long temporal dependencies. (b) corresponds to the intermediate phase, (c) is taken from the final phase where $\sigma$ is at its minimum value (0.5) and weight tuning is more emphasized. Finally, (d) represents the kernel after converting to the discrete form with rounded positions.

datasets that consist of 100k different recordings. The task we consider on SSC and GSC is the top one classification on all 35 different classes (similar to Cramer et al. (2022); Bittar & Garner (2022)), which is more challenging than the original key-word spotting task on 12 classes, proposed in Warden (2018).

For the two spiking datasets, we used spatio-temporal bins to reduce the input dimensions. Input neurons were reduced from 700 to 140 by binning every 5 neurons; as for the temporal dimension, we used a discrete time-step $\Delta t = 10$ ms and a zero right-padding to make sure all recordings in a batch have the same time duration. As for the non-spiking GSC, we used the Mel Spectrogram representation of the waveforms with 140 frequency bins and approximately 100 timesteps to remain consistent to the input sizes used in SSC.

We used a very simple architecture: a feedforward SNN with two or three hidden fully connected layers. Each feedforward layer is implemented using a DCLS module where each synaptic connection is modeled as a 1D temporal convolution with one Gaussian kernel element (as described in Section 3.2), followed by batch normalization, a LIF module (as described in Section 3.1) and dropout. Table 1 lists the values of some hyperparameters used for the three datasets (for more details, refer to the code repository).

Table 1: Network parameters for different datasets

| Dataset | # Hidden Layers | # Hidden size | $\tau$(ms) | Maximum Delay(ms) | Dropout rate |
|---------|-----------------|---------------|-----------|-------------------|--------------|
| SHD | 2 | 256 | 10.05* | 250 | 0.4 |
| SSC/GSC | 2 or 3 | 512 | 15 | 300 | 0.25 |

*We found that a LIF with quasi-instantaneous leak $\tau = 10.05$ (since $\Delta t = 10$) is better than using a Heaviside function for SHD.

The readout layer consists of $n_{\text{classes}}$ LIF neurons with an infinite threshold (where $n_{\text{classes}}$ is 20 or 35 for SHD and SSC/GSC, respectively). Similar to Bittar & Garner (2022), the output $\text{out}_i[t]$ for every neuron $i$ at time $t$ is

$$\text{out}_i[t] = \text{softmax}(u_i^{(r)}[t]) = \frac{e^{u_i^{(r)}[t]}}{\sum_{j=1}^{n_{\text{classes}}} e^{u_j^{(r)}[t]}} \tag{11}$$

where $u_i^{(r)}[t]$ is the membrane potential of neuron $i$ in the readout layer $r$ at time $t$.
The final output of the model after $T$ time-steps is defined as

$$\hat{y}_i = \sum_{t=1}^{T} \text{out}_i[t] \tag{12}$$

We denote the batch size by $N$ and the ground truth by $y$. We calculate the cross-entropy loss for one batch as

$$\mathcal{L} = \frac{1}{N} \sum_{n=1}^{N} -\log(\text{softmax}(\hat{y}_{y_n}[n])) \tag{13}$$

The Adam optimizer (Kingma & Ba, 2017) is used for all models and groups of parameters with base learning rates $lr_w = 0.001$ for synaptic weights and $lr_d = 0.1$ for delays. We used a one-cycle learning rate scheduler (Smith & Topin, 2018) for the weights and cosine annealing (Loshchilov & Hutter, 2017) without restarts for the delays learning rates. Our work is implemented[1] using the PyTorch-based SpikingJelly(Fang et al., 2020; 2023) framework.

## 4.2 RESULTS

We compare our method (DCLS-Delays) in Table 2 to previous works on the SHD, SSC, and GSC-35 (35 denoting the 35 classes harder version) benchmark datasets in terms of accuracy, model size, and whether recurrent connections or delays were used.

The reported accuracy of our method corresponds to the accuracy on the test set using the best-performing model on the validation set. However, since there is no validation set provided for SHD we use the test set as the validation set (similar to Bittar & Garner (2022)). The margins of error are calculated at a 95% confidence level using a t-distribution (we performed ten and five experiments using different random seeds for SHD and SSC/GSC, respectively).

Our method outperforms the previous state-of-the-art accuracy on the three benchmarks (with a significant improvement on SSC and GSC) without using recurrent connections (apart from the self-recurrent connection of the LIF neuron), with a substantially lower number of parameters, and using only vanilla LIF neurons. Other methods that use delays do have a slightly lower number of parameters than we do, yet we outperform them significantly on SHD, while they didn't report any results on the harder benchmarks SSC/GSC. Finally, by increasing the number of hidden layers, we found that the accuracy plateaued after two hidden layers for SHD and three for SSC/GSC. Furthermore, we also evaluated a model (Dense Conv Delay) that uses standard dense convolutions instead of the DCLS ones. This corresponds conceptually to having a fully connected SNN with all possible delay values as multiple synaptic connections between every pair of neurons in successive layers. This led to worse accuracy (partly due to overfitting) than DCLS. The fact that DCLS outperforms a standard dense convolution, although DCLS is more constrained and has fewer parameters, is remarkable.

## 4.3 ABLATION STUDY

In this section, we conduct control experiments aimed at assessing the effectiveness of our delay learning method. The model trained using our full method will be referred to as *Decreasing σ* (specifically, we use the 2L-1KC version), while *Constant σ* will refer to a model where the standard deviation $σ$ is constant and equal to the minimum value of $0.5$ throughout the training. Additionally,

---

[1]Our code is available at: `https://github.com/Thvnvtos/SNN-delays`

Table 2: Classification accuracy on SHD, SSC and GSC-35 datasets

| Dataset | Method | Rec. | Delays | #Params | Top1 Acc. |
|---|---|---|---|---|---|
| **SHD** | EventProp-GeNN (Nowotny et al., 2022) | ✓ | ✗ | N/a | 84.80±1.5% |
| | Cuba-LIF (Dampfhoffer et al., 2022) | ✓ | ✗ | 0.14M | 87.80±1.1% |
| | Adaptive SRNN (Yin et al., 2021) | ✓ | ✗ | N/a | 90.40% |
| | SNN+Delays (Patiño-Saucedo et al., 2023) | ✗ | ✓ | 0.1M | 90.43% |
| | TA-SNN (Yao et al., 2021) | ✗ | ✗ | N/a | 91.08% |
| | STSC-SNN (Yu et al., 2022) | ✗ | ✗ | 2.1M | 92.36% |
| | Adaptive Delays (Sun et al., 2023b) | ✗ | ✓ | 0.1M | 92.45% |
| | DL128-SNN-Dloss (Sun et al., 2023a) | ✗ | ✓ | 0.14M | 92.56% |
| | Dense Conv Delays (ours) | ✗ | ✓ | 2.7M | 93.44% |
| | RadLIF (Bittar & Garner, 2022) | ✓ | ✗ | 3.9M | 94.62% |
| | **DCLS-Delays (2L-1KC)** | ✗ | ✓ | **0.2M** | **95.07±0.24%** |
| **SSC** | Recurrent SNN (Cramer et al., 2022) | ✓ | ✗ | N/a | 50.90 ± 1.1% |
| | Heter. RSNN (Perez-Nieves et al., 2021) | ✓ | ✗ | N/a | 57.30% |
| | SNN-CNN (Sadovsky et al., 2023) | ✗ | ✓ | N/a | 72.03% |
| | Adaptive SRNN (Yin et al., 2021) | ✓ | ✗ | N/a | 74.20% |
| | SpikGRU (Dampfhoffer et al., 2022) | ✓ | ✗ | 0.28M | 77.00±0.4% |
| | RadLIF (Bittar & Garner, 2022) | ✓ | ✗ | 3.9M | 77.40% |
| | Dense Conv Delays 2L (ours) | ✗ | ✓ | 10.9M | 77.86% |
| | Dense Conv Delays 3L (ours) | ✗ | ✓ | 19M | 78.44% |
| | **DCLS-Delays (2L-1KC)** | ✗ | ✓ | **0.7M** | **79.77±0.09%** |
| | **DCLS-Delays (2L-2KC)** | ✗ | ✓ | **1.4M** | **80.16±0.09%** |
| | **DCLS-Delays (3L-1KC)** | ✗ | ✓ | **1.2M** | **80.29±0.06%** |
| | **DCLS-Delays (3L-2KC)** | ✗ | ✓ | **2.5M** | **80.69±0.21%** |
| **GSC-35** | MSAT (He et al., 2023) | ✗ | ✗ | N/a | 87.33% |
| | Dense Conv Delays 2L (ours) | ✗ | ✓ | 10.9M | 92.97% |
| | Dense Conv Delays 3L (ours) | ✗ | ✓ | 19M | 93.19% |
| | RadLIF (Bittar & Garner, 2022) | ✓ | ✗ | 1.2M | 94.51% |
| | **DCLS-Delays (2L-1KC)** | ✗ | ✓ | **0.7M** | **94.91±0.09%** |
| | **DCLS-Delays (2L-2KC)** | ✗ | ✓ | **1.4M** | **95.00±0.06%** |
| | **DCLS-Delays (3L-1KC)** | ✗ | ✓ | **1.2M** | **95.29±0.11%** |
| | **DCLS-Delays (3L-2KC)** | ✗ | ✓ | **2.5M** | **95.35±0.04%** |

nL-mKC stands for a model with n hidden layers and kernel count m, where kernel count denotes the number of non-zero elements in the kernel. "Rec." denotes recurrent connections.

*Fixed random delays* will refer to a model where delays are initialized randomly and not learned, while only weights are learned. Meanwhile, *Decreasing σ - Fixed weights* will refer to a model where the weights are fixed and only delays are learned with a decreasing σ. Finally, *No delays* denotes a standard SNN without delays. To ensure equal parameter counts across all models (for fair comparison), we increased the number of hidden neurons in the *No delays - wider* case, and increased the number of layers instead in the *No delays - deeper* case. Moreover, to make the comparison even fairer, all models have the same initialization for weights and, if required, the same initialization for delays.

We compared the five different models as shown in Figure 4a. The models with delays (whether fixed or learned) significantly outperformed the *No delays* model both on SHD (FC) and SSC (FC); for us, this was an expected outcome given the temporal nature of these benchmarks, as achieving a high accuracy necessitates learning long temporal dependencies. However, we didn't expect the Fixed random delays model to be almost on par with models where delays were trained, with Decreasing σ model only slightly outperforming it.

To explain this, we hypothesized that a random uniformly distributed set of delay positions will likely cover the whole temporal range. This hypothesis is plausible given the fact that the number of synaptic connections vastly outnumbers the total possible discrete delay positions for each kernel. Therefore, as the number of synaptic connections within a layer grows, the necessity of moving

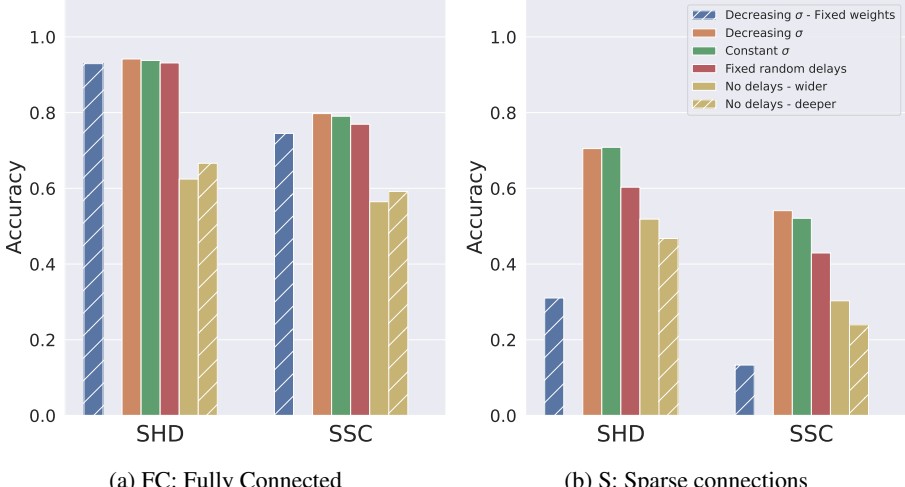

(a) FC: Fully Connected            (b) S: Sparse connections

Figure 4: Barplots of test accuracies on SHD and SSC datasets for different models. With (a): fully connected layers (FC) and (b): sparse synaptic connections (S). Reducing the number of synaptic connections of each neuron to ten for both SHD and SSC.

delay positions away from their initial state diminishes. And only tuning the weights of this set of fixed delays is enough to achieve comparable performance to delay learning.

In order to validate this hypothesis, we conducted a comparison using the same models with a significantly reduced number of synaptic connections. We applied fixed binary masks to the network's synaptic weight parameters. Specifically, for each neuron in the network, we reduced the number of its synaptic connections to ten for both datasets (except for the No delays model, which has more connections to ensure equal parameter counts). This corresponds to 96% sparsity for SHD and 98% sparsity for SSC. With the number of synaptic connections reduced, it is unlikely that the random uniform initialization of delay positions will cover most of the temporal range. Thus, specific long-term dependencies will need to be learned by moving the delays.

The test accuracies corresponding to this control test are shown in Figure 4b. It illustrates the difference in performance between the Fixed random delays model and the Decreasing/Constant $\sigma$ models in the sparse case. This enforces our hypothesis and shows the need to perform this control test for delay learning methods. Furthermore, it also indicates the effectiveness of our method.

In addition, we also tested a model where only the delays are learned while the synaptic weights are fixed (Decreasing $\sigma$ - Fixed weights). It can be seen that learning only the delays gives acceptable results in the fully connected case (in agreement with Grappolini & Subramoney (2023)) but not in the sparse case. To summarize, it is always preferable to learn both weights and delays (and decreasing $\sigma$ helps). If one has to choose, then learning weights is preferable, especially with sparse connectivity.

## 5 CONCLUSION

In this paper, we propose a method for learning delays in feedforward spiking neural networks using dilated convolutions with learnable spacings (DCLS). Every synaptic connection is modeled as a 1D Gaussian kernel centered on the delay position, and DCLS is used to learn the kernel positions (i.e. delays). The standard deviation of the Gaussians is decreased throughout training, such that at the end of training, we obtain a SNN model with one discrete delay per synapse, which could potentially be compatible with neuromorphic implementations. We show that our method outperforms the state-of-the-art in the temporal spiking benchmarks SHD and SSC and the non-spiking benchmark GSC-35 while using fewer parameters than previous proposals. Finally, we also perform a rigorous control test that demonstrates the effectiveness of our delay learning method. Future work will investigate the use of other kernel functions than the Gaussian or applying our method to other network architectures like convolutional networks.

ACKNOWLEDGMENT

This research was supported in part by the Agence Nationale de la Recherche under Grant ANR-20-CE45-0005 BRAIN-Net. This work was granted access to the HPC resources of CALMIP supercomputing center under the allocation 2023-[P22021]. Support from the ANR-3IA Artificial and Natural Intelligence Toulouse Institute is gratefully acknowledged. We also want to thank Wei Fang for developing the SpikingJelly framework that we used in this work.

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

# A APPENDIX

## A.1 SUPPLEMENTARY FIGURE

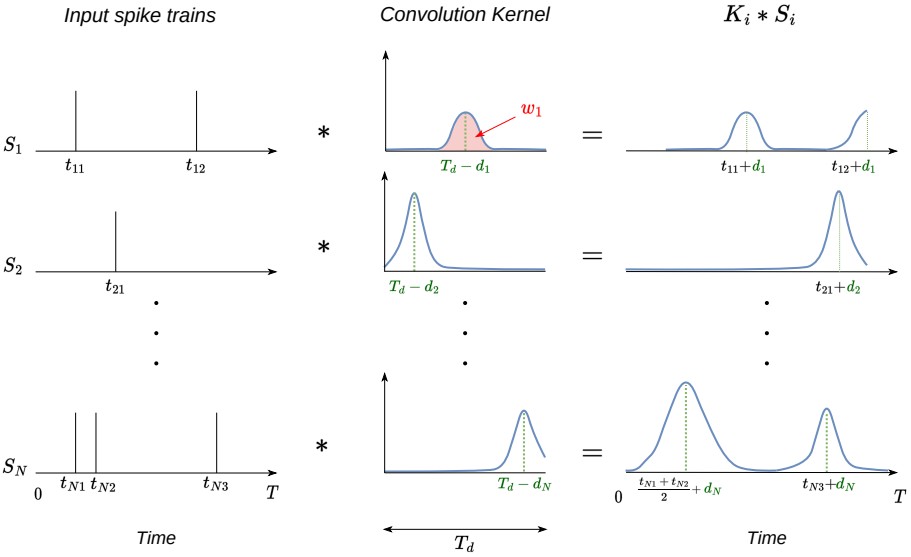

Figure 5: Gaussian convolution kernels for $N$ synaptic connections. The Gaussians are centered on the delay positions, and the area under their curves corresponds to the synaptic weights $w_i$. On the right, we see the delayed spike trains after being convolved with the kernels. (the $-1$ was omitted for figure clarity).

