# OpenReview forum: "Learning Delays in Spiking Neural Networks using Dilated Convolutions with Learnable Spacings"
_ICLR.cc/2024/Conference — ICLR 2024 poster_

### Official Review · Reviewer_7RCo · 2023-10-28

**Soundness:** 3 good
**Presentation:** 3 good
**Contribution:** 2 fair
**Rating:** 6
**Confidence:** 3

**Summary:**

As far as we know, the plastic delays greatly increase the expressivity in SNNs. However, efficient algorithms to learn these delays have been lacking. In this manuscript, the authors propose a new discrete-time algorithm that addresses this issue in deep feedforward SNNs using backpropagation (i.e., offline manner). Then, the kernels contain only a few non-zero weights – one per synapse – whose positions correspond to the delays. Thus, these positions are learned together with the weights using the dilated convolution with learnable spacings (DCLS). The authors show us in a practical way that building deep SNNs can learn together with fixed delays and weights.

**Strengths:**

1. The gap between theory and practice is opened up, especially an efficient fixed delays with weights learning algorithm is designed.
2. The effects of delays can be well explained by visual examples.
3. The anonymous open source code is shared with readers, and the detailed implementation helps inspire readers to build complex and deep SNNs.

**Weaknesses:**

1. The reviewers are very concerned about the innovation of the structure, despite the effort that went into achieving such a particularly efficient discrete-time learning algorithm. The reviewer noted these sentences: "The trick is to simulate delays using temporal
convolutions and to learn them using the recently proposed Dilated Convolution with Learnable Spacings (Khalfaoui-Hassani et al., 2023a;b). In practice, the method is fully integrated with PyTorch and leverages its automatic-differentiation engine." So can we say that structurally this contribution is just a combination that happens to work. This contribution would be improved if the author could further clarify the motivation or give a more solid analysis. After all, a trick feels like an inadequate contribution.
2. Just two datasets with similar statistic information may not seem sufficient, and it would be better if the authors had time to supplement the experiment with a new dataset.

**Questions:**

Please look at the weaknesses.

---

> ### Author Response · Authors · 2023-11-22
>
> The authors would like to thank the reviewer for his constructive questions. In the following, a response to the questions and remarks made by the reviewer:
>
>
> ### Response to Weaknesses
>
> > W1: The reviewers are very concerned about the innovation of the structure, despite the effort that went into achieving such a particularly efficient discrete-time learning algorithm. The reviewer noted these sentences: "The trick is to simulate delays using temporal convolutions and to learn them using the recently proposed Dilated Convolution with Learnable Spacings (Khalfaoui-Hassani et al., 2023a;b). In practice, the method is fully integrated with PyTorch and leverages its automatic-differentiation engine." So can we say that structurally this contribution is just a combination that happens to work. This contribution would be improved if the author could further clarify the motivation or give a more solid analysis. After all, a trick feels like an inadequate contribution.
>
>
> A1: We agree that the main contributions were unclear; we now explain them better at the end of the Introduction, and in the "general response to all reviewers".
>
>
>
>
> > W2: Just two datasets with similar statistic information may not seem sufficient, and it would be better if the authors had time to supplement the experiment with a new dataset.
>
>
> A2: We didn't find other datasets that are a good fit with spiking neural networks in the audio domain. Nevertheless, one of our future directions is to use convolutional spiking neural networks with delays on neuromorphic datasets like DVSGesture.

---

### Official Review · Reviewer_ybas · 2023-10-30

**Soundness:** 3 good
**Presentation:** 3 good
**Contribution:** 4 excellent
**Rating:** 8
**Confidence:** 5

**Summary:**

This paper proposes to study an important problem in spiking neural networks, which involves the explicit incorporation of propagation delays between different neurons in the network. This is an important problem that has been addressed regularly in recent years, and the paper provides a novel and simple solution based on a temporal convolution parameterized by a certain precision. The delays are learned using a variant of the surrogate gradient method, and numerical simulations demonstrate the learning of delays by this method. Experimental results show very good performance on traditional community datasets (in particular, a top score in the leaderboard for the Spiking Heidelberg dataset) and also demonstrate a certain robustness of the method when certain connections are pruned.

**Strengths:**

The paper effectively introduces the problem and motivation and presents the methods clearly. A major strength of the paper is the model's relative mathematical simplicity and its successful performance in supervised classification on two datasets. Promising experimental results indicate that significant energy savings can be achieved with the application of such models to neuromorphic chips by demonstrating the network's robustness when connections are removed.

**Weaknesses:**

The paper's connections with related works are satisfactory; however, it could benefit from presenting neuroscientific evidence on the plasticity of neural delays in biology. Additionally, it lacks a discussion on the relationship between the model's parameters and those observed in biology. For instance, the maximum delays utilized are around 250 milliseconds (300 milliseconds for SSC), while delays used in Izhikevitch's polychronization model are around 20 milliseconds. Furthermore, the paper does not establish any predictions made by the model that can be experimentally observed in biology.

The model has several limitations, such as the use of discrete time, a forward propagation training model, or a limited number of computational layers. However, the presented performance of the network validates the decisions made.

**Questions:**

What is the influence of the meta-parameters on the obtained performance? The influence of the characteristic time of the membrane potential would be interesting to study, as it corresponds to a kind of regularization of spike precision.

Could you comment on the fact that "We found that a LIF with quasi-instantaneous leak τ = 10.05 (since ∆t = 10) is better than using a Heaviside function for SHD." ? Would such a difference matter in biology?

Concerning "We used a one-cycle learning rate scheduler (Smith & Topin, 2018) for the weights and cosine annealing (Loshchilov & Hutter, 2017) without restarts for the delays learning rates. ": Could you comment on your choice of learning rate schedulers? Would different schedulers significantly alter our results? Or does it just improve learning speed?

Minor:
- complete reference for Kingma, for Warden. There seems to be a newer one by Grimaldi for "Learning heterogeneous delays" instead of "Learning hetero-synaptic delays" - plus an additional application paper on motion detection by the same authors.
- spacing: "weights.Hammouamri et al. (2022)"
The LaTeX formatting of the paper is excellent but could be further enhanced. In Figure 1, utilize "N_2", "S_1", and other symbols for clarity. Some citations in the text ("Spike-Element-Wise ResNet Fang et al. (2021b) ", ...) should be enclosed in parentheses, e.g. using `citep`. Text "reset" appearing in equation (1) should be formatted as text, e.g.  using the `\text{}` formatting.

---

> ### Author Response · Authors · 2023-11-22
>
> The authors would like to sincerely thank the reviewer ybas for the high quality of his review. Below we respond to the other questions and remarks made by the reviewer:
>
> ### Response to Weaknesses
>
> > The paper's connections with related works are satisfactory; however, it could benefit from presenting neuroscientific evidence on the plasticity of neural delays in biology. Additionally, it lacks a discussion on the relationship between the model's parameters and those observed in biology. For instance, the maximum delays utilized are around 250 milliseconds (300 milliseconds for SSC), while delays used in Izhikevitch's polychronization model are around 20 milliseconds. Furthermore, the paper does not establish any predictions made by the model that can be experimentally observed in biology.
>
> A1: About delay plasticity in the brain, we cite Bowers 2017, which reviews all the experimental evidence. We do not have the space to review it here, and we think it would be only marginally interesting for the ICLR audience, which is more interested in ML than in neuroscience. For the same reasons, we think that making predictions testable in neuroscience experiments is not required. About the range of delays, we agree that our maximal delays (250-300ms) are longer than what is seen in the brain. Again, this is not a problem for a ML contribution. But if you want to know our opinion, we think that in the brain, many layers are used for sound recognition, so the differences between the fastest path and the longest one across the series of layers accumulate and may reach several hundreds of ms in the end. Here we used only 2 or 3 hidden layers, so the range of delays for each layer had to be unrealistically large.
>
>
>
>
> ### Response to Questions
>
> > Q1: What is the influence of the meta-parameters on the obtained performance? The influence of the characteristic time of the membrane potential would be interesting to study, as it corresponds to a kind of regularization of spike precision.
>
> A2: We agree with this remark, delays and the characteristic time of the membrane potential are very intertwined. Unfortunately, we didn't study thoroughly their relationship in this first work. We want to note that we tried to use learnable membrane potential time constants but it led to slightly worse performance.
>
>
> > Q2: Could you comment on the fact that "We found that a LIF with quasi-instantaneous leak τ = 10.05 (since ∆t = 10) is better than using a Heaviside function for SHD."? Would such a difference matter in biology?
>
> A3: We were also surprised by this experimental result which occurs only on the SHD dataset, we would argue that this difference is only due to numerical reasons and it won't matter in biology.
> Furthermore, we think that since SHD has some examples with lower timesteps compared to SSC/GSC, at the early phases of training when the standard deviation of the delay kernel is high. Delays and membrane potential retention could be redundant and counterproductive.
>
>
> > Q3: Concerning "We used a one-cycle learning rate scheduler (Smith & Topin, 2018) for the weights and cosine annealing (Loshchilov & Hutter, 2017) without restarts for the delays learning rates. ": Could you comment on your choice of learning rate schedulers? Would different schedulers significantly alter our results? Or does it just improve learning speed?
>
> A4: The one-cycle learning rate scheduler seems to work better in general for ML applications with SNNs from our experience. However, the cosine annealing without restarts for delays was specifically chosen due to the standard deviation decreasing strategy we use. Thus, the learning rate is high for the first part of the training when the standard deviation is large and delays can be learned easily, and in the second half of the training, the learning rate is low for delays when the standard deviation is small (see figure 4).
>
>
>
> > Minor
>
> We thank the reviewer greatly for his constructive remarks and feedback.
> we have corrected all the mentioned points, thanks again for your precision!

---

### Official Review · Reviewer_sQsE · 2023-11-01

**Soundness:** 3 good
**Presentation:** 3 good
**Contribution:** 2 fair
**Rating:** 8
**Confidence:** 4

**Summary:**

The paper presents a way to learn synaptic (or axonal) delays in spiking neural networks (SNNs), where the delay of each synapse is realized as a discretized kernel of temporal convolution with a single non-zero element. An evaluation of classification accuracy on three temporal datasets shows that the method works well, with the authors claiming to surpass the state of the art in those datasets.

**Strengths:**

When evaluated within the context of SNNs alone, the paper offers several strengths. Namely, the method is relatively original, the evidence that the method works well is rather convincing, and the impact on SNNs could be significant, given the relative ease of implementation and effectiveness. SNNs themselves interest a growing community.

**Weaknesses:**

The main weakness of the paper is common to many works on SNNs. Specifically, the significance, novelty, potential impact, and experimental validation are limited to the narrow field of SNNs themselves. Very rarely does an SNN paper show its advantages in the broader literature on neural networks, let alone in the real world. The present manuscript too, when evaluated in a broader scope, suffers from the same issues.

More concretely:
- the method is only new for SNNs, but not for neural networks in general.
- the performance is claimed to surpass the state of the art (already in the abstract), but the authors do not actually compare with the true state of the art, including non-spiking networks.
- there is no experimental comparison with standard (i.e. less constrained) temporal convolutions.

Therefore, it is unclear what the true contribution of the work is, beyond a nice conceptual analogy between temporal convolutions and synaptic delays.


Secondary weaknesses:
- Even within spiking networks, the work does not seem to surpass the state of the art, contrary to the authors' claims. In [1], a partly spiking neural network reached 95.6% on the GSC v0.02, where the authors report 95.35% at most. The manuscript does not cite that prior work.
- The paper does not motivate sufficiently the choice of spiking neurons as a model. A paragraph explaining the advantages of SNNs *in comparison with the true state of the art, i.e. ANNs*, supported with citations that demonstrate them measurably, such as energy efficiency, but also rarely in other metrics such as speed of inference and training [1] and even classification accuracy [2]. Any other arguments and citations that the authors can add to support that choice would be useful.
- The authors claim that there is no recurrency in their models, but a leaky integrate-and-fire neuron's leak membrane potential is equivalent to a self-recurrent connection. I understand what the authors mean, but, again in the spirit of appealing to the broader ICLR community and not only to the SNN niche, this should be clarified.

[1] Jeffares et al., Spike-inspired rank coding for fast and accurate recurrent neural networks, ICLR 2022

[2] Moraitis et al., Optimality of short-term synaptic plasticity in modelling certain dynamic environments, arXiv 2021

----------------------------------
EDIT (adding my responses here too, for public visibility):

----------------------------------
The authors' response dedicates a large section to address points that I did not make. To correct the record I must unfortunately reply to that section too, even though it is merely a distraction.

Nowhere did I claim that SNNs are not important or not a legitimate research direction, or that the entire field deserves rejection. I did not dismiss the paper on the basis of it being an SNN. I did point out that some of its weaknesses are frequent in the SNN literature, but pointing that out does not make those weaknesses irrelevant to this specific review. The attempt by the authors to entirely dismiss my review based on how many SNN papers per year are published and how many good reviews the paper received is an attempt to evade my specific criticisms. Worse, the aggressive style of the authors' response, and the misconstrual of my arguments as if they were a personal matter of mine is not helpful.

Again, SNNs can certainly have important advantages, and some SNNs do have them, but a neural network merely being implemented with spiking neurons does not guarantee these benefits. An SNN paper must be evaluated as any other paper, and not merely be accepted as a significant contribution because the network is spiking.

Despite this attempt to discount my comments, I continue my contribution to this process in a separate comment.

----------------------------------

Some important weaknesses remain.

- The key method that the authors used is not new, only its application is.

- The so-far evaluation does not suffice to compare with other works:
(a) Two of the three used datasets have received very little if any attention outside of the SNN literature.
(b) Only feedforward architectures, with only 2 or 3 layers, have been tested.
(c) Only spiking networks have been tested, so it is unclear whether the same results could be achieved, for example, with much smaller (and thus possibly more efficient) non-spiking networks.

- The paper is missing a sufficient motivation of SNNs as a model. A paragraph with the potential benefits of SNNs should be added, citing the previously demonstrated improvements in efficiency, inference speed, and even classification accuracy, but it should also explain that these benefits are not present in all SNNs by default. Examples of such references were given in my original review.


**On the other hand**, the paper now does include a comparison with a more standard method, i.e. conventional temporal convolutions, and it does outperform it. Of course, the work already was a good contribution to the SNN field, but this addition makes it now a relatively convincing demonstration of the power of learned delays more generally, that is a also useful result for the broader ICLR community. Based on these, I am raising my score.

**Questions:**

Could the weaknesses be addressed? Most importantly, could the paper better clarify its significance in the broader field of neural networks? Changes and additions to the text might help address the issues somewhat, but missing experimental evaluations should ideally also be performed, or other measurements of any possible advantage claimed, e.g. number of parameters, energy efficiency etc.

---

> ### Author Response · Authors · 2023-11-22
>
> The authors thank the reviewer sQsE for his review. The main concern of the reviewer is that our paper is mainly interesting for the SNN community. Yet, as the reviewer acknowledges, the SNN community is growing (and supra-linearly see https://www.science.org/doi/full/10.1126/sciadv.adi1480 Fig 28 in Supplementary Material, which plots the number of published SNN papers in top AI conferences, including ICLR). There is enthusiasm because, even if the SNN accuracy does not match (yet?) the one of ANNs, their implementation on neuromorphic chips could be much more power efficient than ANNs on GPUs. Several papers have argued why already. In our opinion, every SNN paper cannot and should not repeat all the arguments.
>
> In addition, some labs and companies have already invested massively to design spiking neuromorphic chips. These actors are not so much interested in the ANN SOTA because they cannot implement ANNs on their chips anyway. But they will be interested in our paper because most of these chips have programmable delays (e.g., Intel Loihi, IBM TrueNorth, Spinnaker, SENECA), and thus could easily implement the SNN we propose for inference.
>
> More generally, we believe that it is up to the Area Chair and the ICLR conference board to decide whether or not SNN papers are welcome at the conference. More explicitly, if our paper is out of scope, then it should have been desk rejected from the beginning. However, this was not the case: the article was not desk rejected. Instead, it was reviewed and received good feedback from half the reviewers. Rejecting the article solely on the basis that the sQsE reviewer is not an SNN enthusiast would be unfair.
>
> Below we respond to the other questions and remarks made by the reviewer:
>
>
> > W1: there is no experimental comparison with standard (i.e. less constrained) temporal convolutions.
>
> A1: We thank the reviewer for this great suggestion! We ran experiments by replacing the DCLS convolutions with standard dense ones, this corresponds conceptually to having a fully connected SNN with all possible delay values as multiple synaptic connections between every pair of neurons in successive layers. This heavy parametrization led to slightly worse accuracy (due to overfitting) than our baseline model while having extensively more parameters. We added these results to Table 2 in section 4.2.
>
>
>
>
> > W2: Even within spiking networks, the work does not seem to surpass the state of the art, contrary to the authors' claims. In [1], a partly spiking neural network reached 95.6\% on the GSC v0.02, where the authors report 95.35\% at most. The manuscript does not cite that prior work.
>
> A2: To the best of our knowledge we achieve state-of-the-art SNN accuracy on SHD, SSC and GSC (PapersWithCode confirms for SHD and SSC: https://paperswithcode.com/sota/audio-classification-on-shd https://paperswithcode.com/sota/audio-classification-on-ssc). [1] reports results on the easier keyword spotting task on GSC v0.02, which considers only 11 classes, while we use the harder task of 35 classes. This is the reason we didn't add it to the result table even though it is an SNN, because it is not the same task.
>
>
>
> > W3: The paper does not motivate sufficiently the choice of spiking neurons as a model. A paragraph explaining the advantages of SNNs in comparison with the true state of the art, i.e. ANNs, supported with citations that demonstrate them measurably, such as energy efficiency, but also rarely in other metrics such as speed of inference and training [1] and even classification accuracy [2]. Any other arguments and citations that the authors can add to support that choice would be useful.
>
>
>
> A3: As explained in the "general answer to all reviewers", there exist many works that specifically focus on measuring energy efficiency and inference speed in comparison to non-spiking ANNs, but it's out of the scope of our paper.
>
>
>
> > W4: The authors claim that there is no recurrency in their models, but a leaky integrate-and-fire neuron's leak membrane potential is equivalent to a self-recurrent connection. I understand what the authors mean, but, again in the spirit of appealing to the broader ICLR community and not only to the SNN niche, this should be clarified.
>
> A4: The reviewer is right. We now say in the manuscript that we don't have recurrent connections *apart from the one of the LIF*.

---

> > ### Comment · Reviewer_sQsE · 2023-12-04
> >
> > Some important weaknesses remain.
> >
> > - The key method that the authors used is not new, only its application is.
> >
> > - The so-far evaluation does not suffice to compare with other works:
> > (a) Two of the three used datasets have received very little if any attention outside of the SNN literature.
> > (b) Only feedforward architectures, with only 2 or 3 layers, have been tested.
> > (c) Only spiking networks have been tested, so it is unclear whether the same results could be achieved, for example, with much smaller (and thus possibly more efficient) non-spiking networks.
> >
> > - The paper is missing a sufficient motivation of SNNs as a model. A paragraph with the potential benefits of SNNs should be added, citing the previously demonstrated improvements in efficiency, inference speed, and even classification accuracy, but it should also explain that these benefits are not present in all SNNs by default. Examples of such references were given in my original review.
> >
> >
> > **On the other hand**, the paper now does include a comparison with a more standard method, i.e. conventional temporal convolutions, and it does outperform it. Of course, the work already was a good contribution to the SNN field, but this addition makes it now a relatively convincing demonstration of the power of learned delays more generally, that is a also useful result for the broader ICLR community. Based on these, I am raising my score.

---

> ### Comment · Reviewer_ybas · 2023-12-03
> **relevance of SNNs**
>
> while there is so far quite a consensus among reviewers about the contribution of this paper, it is quite different in your review, notably by considering the relevance of SNNs in general. I wish to bring forward two aspects:
>
> - it is true that this contribution is quite focused on comparing with other SNNs (the "SNN niche") as the novelty is to extend the representation to the delay domain - something that is not explored in traditional ANNs. yet I would like to stress that it is a novel contribution which may be very useful in future networks in general, in particular when applied to temporal data.
>
> - it seems to me that they indeed each SOTA on the SHD dataset as shown on the official leaderboard (which includes all types of networks).
>
> hope this helps in the final discussion.

---

> > ### Comment · Reviewer_sQsE · 2023-12-04
> >
> > Dear co-reviewer ybas,
> >
> > Thank you for reviving the discussion. I have now replied to the authors in separate comments, and also appended these to the original review to make them publicly visible.
> >
> > Kind regards,
> >
> > Reviewer sQsE

---

> > > ### Comment · Reviewer_ybas · 2023-12-04
> > >
> > > Hi, thanks for your detailed comments co-reviewer sQsE! The paper is clearly missing some comparisons (as many other SNN papers), yet bringing forward some interesting aspects.
> > > best regards, ybas

---

> ### Comment · Reviewer_sQsE · 2023-12-04
>
> The authors' response dedicates a large section to address points that I did not make. To correct the record I must unfortunately reply to that section too, even though it is merely a distraction.
>
> Nowhere did I claim that SNNs are not important or not a legitimate research direction, or that the entire field deserves rejection.
> I did not dismiss the paper on the basis of it being an SNN. I did point out that some of its weaknesses are frequent in the SNN literature, but pointing that out does not make those weaknesses irrelevant to this specific review.
> The attempt by the authors to entirely dismiss my review based on how many SNN papers per year are published and how many good reviews the paper received is an attempt to evade my specific criticisms. Worse, the aggressive style of the authors' response, and the misconstrual of my arguments as if they were a personal matter of mine is not helpful.
>
> Again, SNNs *can* certainly have important advantages, and some SNNs *do* have them, but a neural network merely being implemented with spiking neurons does not guarantee these benefits. An SNN paper must be evaluated as any other paper, and not merely be accepted as a significant contribution because the network is spiking.
>
> Despite this attempt to discount my comments, I continue my contribution to this process in a separate comment.

---

### Official Review · Reviewer_Wbur · 2023-11-01

**Soundness:** 2 fair
**Presentation:** 3 good
**Contribution:** 2 fair
**Rating:** 6
**Confidence:** 4

**Summary:**

In this paper, the authors use the previously published Dilated Convolution with Learnable Spacings (DCLS) method to learn delays in a deep feed-forward spiking neural network using back-propagation. They demonstrate this method on various temporal tasks such as spiking Heidelberg dataset and versions of google speech commands. The authors also demonstrate that learning delays contributes to an increase in performance in sparse networks.

**Strengths:**

- Learning delays, and more generally, using temporal information is a very relevant topic.
- The paper is generally well written and the experiments and setup are clearly described.
- The improvement of performance in networks with fixed sparsity when delays are included is very interesting and this analysis is novel.

**Weaknesses:**

- The novel contribution of this paper over the DCLS paper is not clear. Is it just the evaluation on multiple tasks? It is very important to clarify this aspect.
- The comparison of the model with delays versus no-delays in Sec. 4.3 may not be completely fair: Using more layers (with same number of parameters) for the no-delay network seems more comparable.
- The statement of "Here we show for the first time that delays can be learned together with the weights, using backpropagation, in arbitrarily deep SNNs." is not true. (Shrestha & Orchard 2018) do exactly that.
- Some of the related work are incorrectly cited or not cited:
    - The SLAYER paper (Shrestha & Orchard 2018) does train the delays along with the weights but the authors don't mention it in this context (although it is cited in a different context).
    - dynamically adapting firing thresholds for deep (recurrent) SNNs was first proposed in (Bellec et al. 2018)
    - Spike based transformer references should include SpikeGPT (Rui-Jie et al. 2023) and Spikingformer (Zhou, Chenlin, et al. 2023)

(Shrestha & Orchard 2018) Shrestha, S.B., and Orchard, G. (2018). SLAYER: Spike Layer Error Reassignment in Time. In Advances in Neural Information Processing Systems 31, S. Bengio, H. Wallach, H. Larochelle, K. Grauman, N. Cesa-Bianchi, and R. Garnett, eds. (Curran Associates, Inc.), pp. 1412–1421.

(Bellec et al. 2018) Bellec, G., Salaj, D., Subramoney, A., Legenstein, R., and Maass, W. (2018). Long short-term memory and Learning-to-learn in networks of spiking neurons. In Advances in Neural Information Processing Systems 31, pp. 787–797.

(Rui-Jie et al. 2023) Zhu, Rui-Jie, Qihang Zhao, and Jason K. Eshraghian. "Spikegpt: Generative pre-trained language model with spiking neural networks." arXiv preprint arXiv:2302.13939 (2023).

(Zhou, Chenlin, et al. 2023) Zhou, Chenlin, et al. "Spikingformer: Spike-driven Residual Learning for Transformer-based Spiking Neural Network." arXiv preprint arXiv:2304.11954 (2023).

**Questions:**

## Suggestions:

- The DVS gesture recognition dataset, due to its event-based nature, might have been a really good fit for a method that learns delays.
- Since delays use temporal information, it might have made more sense to use a loss function that made use of this (for e.g. time-to-first-spike loss)

### Minor:
- Acronyms for task names are not explained in the results section

---

> ### Author Response · Authors · 2023-11-22
>
> The authors thank the reviewer Wbur for his review. The following is a response to the questions and remarks made by the reviewer:
>
>
> ### Response to Weaknesses
>
> > W1: The novel contribution of this paper over the DCLS paper is not clear. Is it just the evaluation on multiple tasks? It is very important to clarify this aspect.
>
> A1: As we explained in the "general answer to all reviewers", the idea of modeling fully connected SNNs with delays using temporal convolutions and learning jointly the weights and delays doesn't come trivially from the DCLS paper (Khalfaoui et al 2023), which presents a general method for n-dimensional convolutions and used primarily for the spatial domain. Following this remark we modified the last part of the Introduction to explain it better.
>
>
>
> > W2: The comparison of the model with delays versus no-delays in Sec. 4.3 may not be completely fair: Using more layers (with same number of parameters) for the no-delay network seems more comparable.
>
> A2: Following this remark, we ran new experiments for the no-delays SNN, using more layers while remaining at the same number of parameters by decreasing the number of hidden neurons in each layer, the results for these runs are grouped in the table below, we also added them to the figure 5 in section 4.3.
>
> | Model             | SHD        | SSC     |
> | :---              |:----:      | :----:  |
> | 2 Layers          | 62.45\%    | 56.47\% |
> | 3 Layers          | 66.62\%    | 59.18\% |
> | ================= | =======    | ======= |
> | 2 Layers - sparse | 51.87\%    | 30.29\% |
> | 3 Layers - sparse | 46.75\%    | 23.95\% |
>
>
> In the non-sparse case, adding more layers improves the performance by approximately 4\% for both datasets. While in the case of sparse synaptic connections, it actually makes the performance much worse, due to the fact that the effect of sparsity is more important when having less hidden neurons. In general, the main claims of the ablation section remain the same.
>
> Most importantly, both networks do much worse than the SNNs with delays. So the conclusion of the paper is unchanged.
>
>
>
> > W3: The statement of "Here we show for the first time that delays can be learned together with the weights, using backpropagation, in arbitrarily deep SNNs." is not true. (Shrestha & Orchard 2018) do exactly that.
>
> A3: Following this important remark, this claim was removed from the paper. What we meant is that we were the first ones not to use a finite element approximation to calculate the gradient of the delay (as explained in section 2.2). However, (Shrestha & Orchard 2018) can do that too with the condition of using an SRM neuron model.
>
>
>
>
>
> > W4: Some of the related work are incorrectly cited or not cited
>
>
> A4: Following this important remark we cited all the missing works. And especially for (Shrestha & Orchard 2018), the explanation in the previous answer A3, was added to section 2.2.
>
>
> ### Response to Questions
>
> > Suggestion1: The DVS gesture recognition dataset, due to its event-based nature, might have been a really good fit for a method that learns delays.
>
> A5: We agree, but visual tasks usually require convolutions in the spatial domain. Our work focuses on simple fully connected networks, and extending it to convolutional spiking neural networks is one of our future directions. We started with tasks that are very dependent on temporal pattern detection since we think delays help most in these types of tasks.
>
>
>
>
>
> > Suggestion2: Since delays use temporal information, it might have made more sense to use a loss function that made use of this (for e.g. time-to-first-spike loss)
>
>
> A6: Following this remark, we tried to use a loss function based on the time-to-first-spike (TTFS). The resulting accuracy was above the chance level, but very poor (around 8\% on SHD). We think TTFS is ill-suited for sequence classification like here, because TTFS encourages early spikes, which, due to causality, ignore the end of the sequences.
>
>
>
> > Minor: Acronyms for task names are not explained in the results section
>
> A7: Thank you for this suggestion, we added the full names when the acronyms are used for the first time, i.e. at the beginning of the section 4.1 - experimental setup.

---

> > ### Comment · Reviewer_Wbur · 2023-11-22
> >
> > Thank you for the clarifications and updates to the paper. I'm convinced that this paper has additional contributions over the DCLS paper, but I still feel they are somewhat incremental -- application to SNNs, different applications, and small changes in architecture and intention.
> > I thank the authors for the additional ablations, and the other fixes, which improves the paper.
> > Overall, I'm willing to increase my score to 6.

---

### Author Response · Authors · 2023-11-22
**General answer to reviewers**

We sincerely thank all the reviewers for their valuable comments. In light of these comments, we have run new experiments and modified the article. We think that the quality of our paper has been much improved.


Some reviewers were concerned about novelty.
The DCLS method has been published before (Khalfaoui et al 2023).
Yet the present paper goes way beyond the application of the method to another dataset.

There are major differences between the two papers:

* Khalfaoui et al. 2023 used images, not sounds.

* Khalfaoui et al. 2023 used DCLS2d in the spatial domain, not DCLS1d in the temporal domain.

* Khalfaoui et al. 2023 presented DCLS as a "method to increase the RF size without increasing the number of parameters", not to learn delays.

* Most importantly, Khalfaoui et al. 2023 did not use spikes. SNNs are not even mentioned in that paper, nor delays!
As reviewer sQsE acknowledged, our paper presents a "nice conceptual analogy between temporal convolutions and synaptic delays".  This analogy was not mentioned in Khalfaoui et al 2023, and does not follow trivially from it.
Thanks to this analogy, we show that DCLS can be used to learn delays in SNNs and that doing so improves the SNN SOTA on several audio datasets.

At the end of the Introduction of this revised version, we explain better the novelty of the paper.


**Below we respond to each of the reviewers's points**

---

### Meta-Review · Area_Chair_YSEt · 2023-12-05

**Metareview:**

The authors propose to learn delays with weights in SNNs.  The delays are modeled using 1D convolutions across time in the Dilated Convolution with Learnable Spacings (DCLS) framework such that backprop can be naturally applied.  Experimental results on various datasets show strong performance of the proposed SNNs with learnable delays.  Overall, this is an interesting paper that is well motivated and well written. The experiments are well designed and controlled. The results are convincing.   Although the DCLS framework itself is not novel,  the way of converting delays in SNNs with 1D convolution and making it learnable using backprop appears to be original.  The rebuttal has cleared most of the concerns raised by the reviewers with clarifications and added experiments.  This work may have its value to the SNN community.  After the discussion, all reviewers are supportive of accepting it.

**Justification For Why Not Higher Score:**

The DCLS framework itself is not novel. There is also related work on learning delays and weights together in literature.

**Justification For Why Not Lower Score:**

This is an interesting work in general.  The way of converting delays in SNNs to a 1D convolution to learn with backprop is original. The results are also very strong.

---

### Decision · Program_Chairs · 2024-01-16

Accept (poster)